# Do Common Beans (*Phaseolus vulgaris* L.) Promote Good Health in Humans? A Systematic Review and Meta-Analysis of Clinical and Randomized Controlled Trials

**DOI:** 10.3390/nu13113701

**Published:** 2021-10-21

**Authors:** Eileen Bogweh Nchanji, Odhiambo Collins Ageyo

**Affiliations:** International Centre for Tropical Agriculture, Nairobi P.O. Box 823-00621, Kenya; collinsaraf@gmail.com

**Keywords:** common beans, cardiovascular diseases, diabetes mellitus, obesity, cancers, human trials, health

## Abstract

The common bean is a nutrient-dense food empirically known to have beneficial effects on human health. Many studies have looked at the effects of “pulses” on different health issues, providing general overviews of the importance of each pulse in health studies. This study systematically reviews and provides meta-analyses of the effect of bean extract as a supplement or whole bean on four health issues (cardiovascular diseases, diabetes mellitus, obesity, and cancers) from a dissection of clinical and randomized controlled trials using human subjects. A digital search in PubMed and Google Scholar^TM^ resulted in 340 articles, with only 23 peer-reviewed articles matching our inclusion criteria. Findings indicated that common beans reduced LDL cholesterol by 19 percent, risk of cardiovascular disease (CVD) by 11 percent, and coronary heart disease (CHD) by 22 percent. Besides this, we noted variances in the literature on cancer findings, with some authors stating it reduced the proliferation of some kinds of tumor cells and reduced the growth of polyps, while others did not specifically examine cancers but the predisposing factors alone. However, diabetes studies indicated that the postprandial glucose level at the peak of 60 min for common bean consumers was low (mean difference = −2.01; 95% CI [−4.6, −0.63]), but the difference between the treated and control was not significant, and there was a high level of heterogeneity among studies (I^2^ = 98%). Only obesity studies indicated a significantly high level of weight gain among control groups (mean difference = 1.62; 95% CI [0.37, 2.86]). There is a need for additional clinical trials using a standardized measure to indicate the real effect of the common bean on health.

## 1. Introduction

Globally, pulses are regarded as nutritional powerhouses and an alternative component of healthy diets among poor households [1]. This is especially true for households in Sub-Sahara Africa (SSA), where it forms the largest part of the diet, with weekly consumptions averaged at 4.2 times a week in some countries, such as Uganda [2]. The global per capita intake of pulses is about 21 g per day, with SSA averaging 33 g per capita per day [3]. This can be as high as 107 g per meal per person for common bean (*Phaseolus vulgaris* L.) consumers in Uganda, averaging an annual of 22.41 kg/person [2]. Common beans were once considered a “poor man’s meat” across many countries in Sub-Saharan Africa [4,5], because they were not being consumed by the rich. They are the most produced crops, and are second only to maize in some countries, such as Kenya [6].

In Rwanda, for example, high-iron bean (HIB) varieties developed by the Pan-Africa Bean Research Alliance (PABRA) are grown by 20 percent of the farmers and consumed by 15 percent of the population (or 1.8 million people), in response to the need for fighting hidden hunger and malnutrition [7]. Several studies in Rwanda [8,9,10] demonstrated that consumption of HIB had cognitive benefits for college students, increased physical work efficiency in women, and increased iron status in women after 128 days, respectively. However, there exist no long-term studies or national-level datasets on the effectiveness of bio-fortification in increasing iron in common beans with respect to SSA.

Common beans are also lauded as ecologically sustainable protein sources compared to animal-based ones [11]. Additionally, they have a unique nutrient-rich profile, especially their high protein content, ranging between 17 and 30 percent dry weight [12]. The high insoluble fiber in beans also provides slow-digesting carbohydrates and micronutrients (iron and zinc) that reduce postprandial glucose release, thereby benefiting people with diabetes. Apart from health, beans are also beneficial economically, especially to women in SSA, who are disproportionately involved in its production, processing, marketing, and cooking [13].

Contemporary developments have given rise to rejuvenated interests in epidemiological, clinical, and randomized controlled trial studies that investigate common bean consumption with respect to health, especially their role in reducing the risks of chronic cancers, cardiovascular diseases (CVD), diabetes mellitus, and obesity [1,14]. This interest is pegged on the nutritional capacity of beans given their relatively low cooked-serving energy density, low-fat content, and high iron and zinc, among other minerals (1.3 kcal/g) [15]. The 50–65 percent carbohydrate content of common beans is slowly digested, further lowering the glycemic index (GI) [16]. Furthermore, common beans provide 23–32 g of fiber per 100 g dry weight, of which 20–28 g is insoluble, highlighting their role as fiber-rich foods, which can lower the risk of heart disease, stroke, diabetes, and colorectal cancer [17]. 

Despite the interest in common bean studies, a review of meta-analysis studies that have delved into pulses in general reveals some interesting study gaps. First, according to [18], there is a huge gap in studies that use human subjects to examine the effect of pulses on health. This current study only includes studies that only used human subjects, which may be desirable for dietary recommendations for humans. Secondly, meta-analysis studies such as Ferreira et al. [1] and Marventano et al. [19] generally reported on “pulses” and not specifically on common beans. Thirdly, a mixed use of human subjects and mice/rats presents a challenge in pinpointing the effect of common beans on human health. As mentioned above, all mice studies were therefore excluded. Furthermore, studies such as Onakpoya et al. [20], Padhi and Ramdath [21], Sievenpiper et al. [22], and Zhu et al. [23] conducted meta-analyses on pulses, focusing on specific health issues such as CVD and colorectal cancer; prior studies have seldom combined all health issues that pulses may alleviate. This means such studies present a one-sided story that may limit our understanding of the full range of benefits of common beans on other health issues. In the current study, we specifically focus on common beans and include four main health issues (cancer, CVD, obesity, and diabetes) that have not been combined in other meta-analyses. We also differ from studies using mice, instead reporting on clinical and RCT trials using human subjects. This, to the best of our knowledge, has previously not been presented or analyzed.

## 2. Materials and Methods

### 2.1. Search Strategy

We used the Preferred Reporting Items for Systematic reviews and Meta-Analyses (PRISMA) criteria for the literature review in this study [24]. We first searched the indexed studies found on Google Scholar with a powerful search query as follows: intext: “common beans” AND “health” AND “randomized controlled trial” -fruit -herbal -mice filetype: pdf. The ‘minuses’ excluded fruit and herbal fusions and studies that used mice. Each time, we iteratively replaced “health” with either of the four health issues and restricted results to be from 1995 to 2021. The searches were conducted between the dates of 15 March and 15 April 2021. We also searched PubMed with the terms “common beans” and “health” and applied filters for clinical trials and randomized controlled trials; in each specific search on the health issue, the term “health” was replaced with “cancer”, “CVD”, “obesity”, and “diabetes.” 

### 2.2. Study Selection and Inclusion and Exclusion Criteria

Peer-reviewed articles from reputable journals were included based on the following criteria: (1) used human subjects and not mice/rats; (2) used common beans as a whole grain or extract as a sole treatment or as part of a treatment, not “pulses” in general; (3) written in English; (4) used RCT or clinical trials; (5) not a systemic review or meta-analysis, editorial, expert opinion, review, or instructive article. Studies that used meta-analysis were excluded because they used specific criteria for selection, and 90 percent of them concentrated on a single issue, e.g., CVD. Blogs, web pages, opinion pieces, reports, and magazine articles were excluded due to a lack of scientific rigor in them. Studies that combined pulses (for example, common beans and lentils) were excluded because the effect of a particular grain on human health would be challenging to evaluate in such a situation. A total of 24 studies were included in our final analysis, as shown in Figure 1 below.

### 2.3. Data Extraction

From each paper, data relating to demographic characteristics of the participants such as author(s) name, year of publication, total participants disaggregated by sex and treatment (males vs. females and treated vs. control group), body mass index (BMI) in kg/m^2^, and age bracket of the participants were first extracted. Furthermore, to understand the length of the intervention, data on weeks were evaluated, and the daily quantity of beans in grams (dry weight) per day for whole cooked beans and milligrams per day for bean extracts (wet weight) were extracted next. The mean difference or percentage difference in outcomes between the control and treated group was then extracted. The bean extracts are in milligrams, as their conversion to grams/day (wet weight) would make them so small and incomparable to the whole bean group. Data were also extracted in terms of the health issues addressed in this paper, i.e., diabetes, CVD, obesity, and cancer. The common bean variety used in the study was also extracted. Finally, the study summary or policy direction was extracted; the summaries are presented in Table 1 under summary statistics.

### 2.4. Empirical Estimations and Statistical Tests

In the current study, we use a mixed-method approach and combine descriptive statistics, forest plot analysis, and random-effects meta-regression. Descriptive statistics are used to present summaries in Table 1, providing basic information about the variables used in the datasets and indicating the potential relationships between the variables used in the study. Forest plots are necessary for the visualization of the mean differences and potential influences (measured by weights) that each study has in the meta-analysis. Various studies have used forest plots for meta-analysis [19,23,24,45].However, in this case, we additionally combined the forest plots with meta-regressions to highlight the effect of the ‘potential effect modifiers’ or covariates on the mean differences [45].The influence on the relationship and the residual heterogeneity among intervention effects that are seldom captured by the explanatory variables are also captured. The first instance is the obvious OLS regression, in which an estimation of the relationship between the outcome and covariates is estimated as follows: (1)$${\hat{y}}_{i}=\beta_{0}+\beta_{1}x_{i}$$
where $y_{i}$ is the mean difference of study outcomes, $\beta_{0}$ and $\beta_{1}$ are the coeffeicients, and $x_{i}$ is a covariate. This can be extended to meta regression, where the outcome is the observed effect size $\theta^{k}$ of study k. Thus,
(2)$$\theta_{k}=\beta_{0}+\beta_{1}x_{k}+\varepsilon_{k}+\zeta_{k}$$
where the two types of independent errors are $\varepsilon_{k}$ (the sampling error showing how the effect size of the studies vary from the true effect) and $\zeta_{k}$ (denoting that the true effect size is a subsample of the distribution of effect sizes). The coefficients can be interpreted as unit changes in covariates as the outcome changes. Following [46], we used the Sidik–Jonkman method and applied the Knapp–Hartung error adjustment to present robust coefficients. This is because the widely used DerSimonian–Laird (DL) method assumes that different studies estimate different, yet related, intervention effects, which has some fundamental weaknesses [47]. For example, while treatment and testing inefficaciousness should have an error rate of 5%, in the DL method, the error rate is usually substantially higher in smaller studies with a smaller sample size like ours. While there are alternatives to meta-regression, studies have consistently demonstrated that using the Knapp–Hartung method gives better adjustment to the significant levels and higher power rates [48].

## 3. Results

### 3.1. Summary Findings of Studies

Table 1 shows studies and other control covariates of interest that authors have presented in different papers. The studies ranged from 4 to 8 weeks, with cancer studies taking the least time, averaging 4 weeks (a month). Participants were given either bean extracts or whole grains in grams per day in terms of the interventions. On average, 195.82 g/day (dry weight) of cooked whole beans was given to participants. This may be equivalent to ¾ of a cup of cooked common beans per day, except for obesity studies, which used 1 ¾ cups of cooked beans per day. Both men and women were used in the studies; however, almost none of the papers considered gender disaggregation of outcomes.

We also looked at the specific common bean varieties used for the studies. It emerged that 21 percent of studies seemed to prefer navy, red, and pinto beans for their studies (calculations not shown in the table). There were cases where different varieties were combined, and in such cases, navy beans seemed to be used as a priority, followed by pinto beans. In the United Kingdom and the United States, where most of these studies took place, navy beans or haricot beans are particularly popular [49]. White beans are also popular and have high phosphatidylserine levels, which is beneficial to most consumers (ibid). Specifically, CVD studies preferred navy, pinto, or red bean varieties, or a combination thereof. Cancer studies used navy, kidney, or pinto, or a combination thereof. Diabetes studies were diverse in terms of bean varieties used, while obesity studies seemed to prefer white kidney beans.

#### 3.1.1. Common Beans and Cardiovascular Diseases (CVD)

For studies that looked at common beans and their effects on CVD, three papers were particularly vital. First, [25] used the largest sample size for people with no CVD at the beginning and registered 3680 incidents of CVD after a 19-year follow-up. The authors found that those who consumed dry beans such as pinto and red beans had an 11 percent lower rate of CVD. Furthermore, [27] found that 180 g/day (dry weight) of bean consumption reduced the mean serum total cholesterol by 19 percent. However, Cryne et al. [26], having administered ½ cup of beans per day, failed to find any significant difference in serum lipids, homocysteine, or glycemic parameters, meaning that beans are not biomarkers of CVD risk. 

a.Common beans and cancers

The five studies evaluated for cancers concentrated on different types, especially colorectal and breast cancer and other tumors associated with cancers (Table 1). First, Hartman et al. [28] administered 251 g/day (dry weight) of whole beans for men characterized for colorectal adenomas. The authors found that soluble tumor necrosis factor-alpha receptor I and II (sTNFRI/II) concentrations declined marginally during the legume diet period (23.8%; *p* = 0.060). This implies that TNFRII directly promoting the proliferation of some kinds of tumor cells reduced with bean consumption. On the other hand, Perera et al. [29] found elevated serum pipecolic acid and S-methyl cysteine (which combats the growth of polyps, i.e., abnormal cell growth that can eventually become malignant) after dry bean consumption. Using 35 g/day (dry weight) of navy beans, Baxter et al. [45] found that relative stool abundance of ophthalmate increased 5.25-fold, meaning that cancer control mechanisms such as detoxification of xenobiotics, antioxidant defense, proliferation, and apoptosis were enhanced by common bean consumption. Borresen et al. [31] found serum amyloid A (SAA) levels after bean consumption at week 4 to be high enough to be associated with colorectal cancer chemoprevention.

b.Common beans and diabetes

Seven studies reported postprandial glucose and insulin levels within 30, 60, 90, and 120 min after bean consumption, as shown in Table 1. The mean peak glucose level for most studies was reported at 60 min; therefore, we derived mean differences in the fasting glucose level at that number of minutes across all eight studies. First, Nilsson et al. [33] reported that administration of 101 g/day (dry weight) of brown beans lowered blood glucose (215%, *p* < 0.01) and insulin (216%, *p* < 0.05) at the peak levels. The authors concluded that brown beans had some colonic substrates that may prevent obesity and metabolic syndrome through glucose regulation. On the other hand, Olmedilla-Alonso et al. [34] found the maximum glucose level after 60 min to be almost the same for the two varieties of beans used, where Almonga had 149.8 ± 28.8 mgdL^−1^ and Curruquilla had 145.3 ± 22.4 mgdL^−1^, while the maximum insulin concentration for Almonga was 33.2 ± 34.7 µlU/mL. The insulin concentrations were 2.48 times lower than the control group (−0.28), which had white bread. Using a new, standardized, and purified common bean extract (PVE), Spadafranca et al. [35] found a significantly lower percentage (15.4) of blood glucose compared to the placebo group. The authors also found insulin increased in percentage less after PVE supplementation than after placebo (+981 (SEM 115) vs. 1325 (SEM 240). On the other hand, Reverri et al. [36] found that after 60 min, postprandial blood glucose peaks were lower by 0.2 mmol/L for those who consumed black beans, and reduced postprandial insulin concentrations were also reported. Kazemi et al. [38]) found the pulse-based group to have a greater reduction in total area under the curve for insulin response (mean change −121.0 ± 229.9 vs. −27.4 ± 110.2 µIU/mL × min; *p* = 0.05). The authors concluded that pulse-based diets reduce cardio-metabolic disease (e.g., diabetes mellitus) risk factors in women with polycystic ovary syndrome (PCOS). Thompson [39] reported that net change glucose responses were significantly lower for the pinto, black, and dark red kidney bean and rice meals compared to control at 90, 120, and 150 min post-treatment (*p* < 0.05). The author recommended a bean and rice mixture for controlling Type II diabetes. However, [37] found no significant differences in glucose levels between the treated and the placebo group.

c.Common beans and obesity

Seven obesity studies were reviewed. The mean weight loss/gain due to consumption of common bean grain or extract was derived. Udani and Singh (2007) found that, after consuming bean extract, the treatment group lost 6.0 lbs (or 2.7 kg) (*p* = 0.0002) and 2.2 inches in waist size (*p* = 0.0050), while the placebo group lost 4.7 lbs (or 2.1 kg) compared to the baseline.

On the other hand, Wang et al. [14] found that those who consumed PVE had an average weight loss of 2.24 kg (2.7%), compared to only 0.29 kg (0.3%) for the placebo group. Birketvedt et al. (2004) found that the supplement group had a significant reduction in body weight of −3.2 ± 3.4 kg, whereas the placebo group only had a −0.2 ± 2.3 kg reduction. In another study by Grube et al. [42] with 57 obese subjects, the treatment group had a mean weight loss of 3.02 ± 62.97 kg, while the placebo group lost a mean of 1.22 ± 62.36 kg (*p* = 0.027). Maruyama et al. [43] found that the 10 individuals who took the control juice gained 0.6 kg in weight compared to the 0.1 kg weight loss of the group that took the concentrated adzuki juice. Celleno et al. [44] found that the body weight (kg) of those who took the bean extract decreased by −2.93 ±1.6 kg, whereas the placebo group lost −0.35 ± 0.38 kg (*p* < 0.001). The final study evaluated by Winham and Hutchins [37] found no significant differences in weight change over the two treatment periods for the treated and the placebo group. However, the authors reported a decrease in serum low-density lipoprotein cholesterol (LDL-C) by −5.4 ± 2.3 percent, possibly indicating that beans reduce LDL-cholesterol (which can lead to obesity).

#### 3.1.2. Empirical Estimations

a.Forest plot analysis (obesity and diabetes studies)

Figure 2 shows a forest plot. Column 1 shows the studies that were included, while columns 2 and 3 show the mean differences and standard deviations of postprandial glucose level after 60 min for the diabetes studies. The mean body weight reduction (as percentages) for the participants after consuming whole beans or their extracts are also presented. The forest plot also shows the confidence intervals of several studies represented by the whiskers (e.g., 95% CI), while the boxes represent the weight given to the study. In addition, weights (as percentages) indicating the contribution of an individual study on the pooled result are presented, where studies with bigger sample sizes and a narrower confidence interval (CI) correspond to a higher percentage weight and a larger box.

The horizontal red “line of null effect” for these studies indicates that most of the studies crossed the line, meaning they had little to no effect and do not illustrate a statistically significant result. The diamond shape indicates the general significant effect of the studies combined.

The diabetes studies seem to indicate a non-significant reduction in glucose levels (the horizontal tip of the diamond crosses the vertical line). It is worth noting that most diabetes studies show that common bean consumption (whether extract or whole grain) effectively reduces postprandial glucose levels among the treated group because the CIs are entirely on the negative side and favor the treated group. Overall, the diabetes studies’ results indicate a medium treatment effect (mean difference = −2.01), 95% CI [−4.6, −0.63]). The Q statistic (Q = 70.06, df = 6) indicated that across the diabetes studies, the effect sizes differed significantly. While the I^2^ of 98% suggests a high level of heterogeneity, and given the rule of thumb that less than 50% is desirable, we may conclude that the interventions or exposure were not consistent across all the studies, or a random-effects meta-regression may be needed to discern the effects of individual covariates. The high heterogeneity may be attributable to the small sample size of the studies we found, the type of common beans used, and perhaps the researchers’ own biases and problems with data collection. The observed variances can also be attributable to differences at the study level. The T^2^ of 10.62 suggests a medium amount of absolute dispersion. Following the suggestions of Baker et al. [45], higher heterogeneity (variation in results across studies) may require a meta-regression to see the effect of each covariate on the outcome. Given the small number studies (low sample size), it is our opinion that conclusions from such regressions would be useful but not generalizable. The studies conducted by Olmedilla-Alonso et al. [34] and Reverri et al. [36] seem to have the largest impact, due to their narrow CI.

On the other hand, the obesity studies seem to favor the control group, but in the true sense, it suggests that they had the largest weight gain with medium treatment effect (mean difference = 1.62), 95% CI [0.37–2.86]). The study desired to observe reduced weight among the participants. However, we cautiously present the mean difference, as it may not represent the true mean of the reduction in body weight due to the high heterogeneity (I^2^ = 94%). Instead, following Baker et al. (2009), the range of mean differences for the studies whose means fall within the 0.37 and 2.86 CI may offer insight into the effect of common beans on weight loss. The CI indicates that the true mean value that other studies might find for the effect of the common bean on weight gain/loss lies between 0.37 and 2.86. The weight loss is not significant but suggests that common beans may be used to control obesity. The Q-statistic (Q = 348.27, df = 6, *p* < 0.001) indicates that across the studies on obesity, the effect sizes differ significantly. The first three studies by Celleno et al. [44], Grube et al. [42], and Wang et al. [14] are given weights as 8.05, 7.94, and 8.07, respectively. The mean difference of weight in favor of the control group is between 1.8 kg and 3.4 kg. Basically, it may be assumed that consuming common bean extract (for example, capsules made from beans or concentrated bean juices) for 4–12 weeks has a weight loss effect of between 1.8 and 3.4 kg. However, the difference between the treated and the control group is not significant. We may attribute the non-significant difference to non-standardized measures of common beans (extracts vs. whole beans).

b.Random-effects meta-regression

The meta-regression was run with the mean differences in outcomes (postprandial glucose level after 60 min and mean weight loss) as the dependent variables and age, weeks of the study, and quantity of beans consumed as the potential predictors.

As indicated in Table 2, while age may be strongly prognostic, we found evidence that an increase in age by one year was associated with the likelihood of increased obesity by up to 0.19 kg. However, it had a negative effect on diabetes, although the difference was not significant. This may be associated with the fact that the mean age for diabetics was 38 years (younger populations), and studies have found the onset of diabetes to be about 45 years [49]. Perhaps the authors used younger populations that may not reflect the potential effect of age on diabetes using common bean as a control. On the other hand, the number of weeks of the study seemed to significantly reduce the mean outcomes for both obesity and diabetes. This is counter-intuitive and may be explained by the fact that most of the studies had no follow-ups or repeated the treatment after some time to observe if treatment and no other factors significantly affected the outcomes. After accounting for age, weeks, and quantity of common bean consumed, we found the remaining between-study residual heterogeneity to be roughly I^2^ = 56.78% for diabetes studies and I^2^ = 38.37% for obesity studies.

## 4. Discussion and Conclusions

In our analysis of 23 papers linking common bean consumption to four health outcomes (cardiovascular diseases, diabetes mellitus, obesity, and cancers), some interesting outcomes relating to regular bean consumption were observed. First, common beans have been demonstrated to reduce LDL cholesterol by 19 percent and lower CVD rates by 11 percent. Perhaps the work of Bazzano et al. [25] using pinto and red bean varieties may be reliable because it demonstrated a 22 percent reduction of coronary heart disease (CHD) and 11 percent reduction of cardiovascular disease (CVD). However, the remaining studies differed in terms of the percentage reduction, and the figures failed to be as near as possible to each other, as should happen in RCTs. This may not be surprising given the different bean varieties used (differing in terms of their nutrient content), the agro-ecological zones, the subjects evaluated, and more importantly, the form of common bean administered (extracted or whole grain). Evidently, common beans may only be an option, but not the ultimate solution, for reducing bad fats that directly lead to fat accumulation, which is a risk factor for heart disease. Therefore, common beans can be recommended as a component in diets but may need to be standardized in terms of the daily dose to give people within a given weight range and, most importantly, the variety of common beans to use. In our case, pinto and navy beans may be recommended because they seem to have potential for reducing CVD.

On the other hand, cancer studies were a bit haphazard with respect to studying how common beans affect colorectal adenomas and the proliferation of some kinds of tumor cells and how beans reduce the growth of polyps. Just like in CVD, there was a general lack of specificity for the type of cancer targeted. Even though colorectal and breast cancer received a fair share of studies, other more dangerous types of cancers (such as throat cancer) were not included. Perhaps the work of Ombra et al. (2016) that analyzed extracted cancer cells and indicated that 10 g of bean consumption may be capable of inhibiting the proliferation of human epithelial colorectal adenocarcinoma (Caco-2) cells, breast cancer cells, and non-small-cell lung cancer (NSCLC) cells should set an example for works relating to common beans and health. However, since it was not conducted in RCT or clinical trials, we could not include it. To date, cancer studies may be concluded to be underdeveloped and may not demonstrate whether common beans reduce the likelihood of getting cancers. Further research is required.

Diabetes and obesity studies seem to have more developed discourses regarding common bean and health research. While various studies have demonstrated that postprandial glucose levels at the peak of 60 min were lower for bean consumers, the percentage reduction was most of the time not significantly different. For example, while Spadafranca et al. [35] found a significantly lower percentage (15.4 percent) of blood glucose, Winham and Hutchins [37] found no significant differences. Understandably, the bean variety used, number of subjects, and type of bean consumed (extract or whole) may have influenced the study outcomes, but the 15 percentage difference clearly demonstrates that the non-standardized approach leads to counterintuitive results. Nevertheless, the forest plot (Figure 1) shows that common bean consumption clearly favors the treated group, although there is no significant difference. This may mean that common bean consumption may not be used as a stand-alone recommended diet for glucose control, but as an option for more controlled glucose release after meals by the body. This is demonstrated in most studies, especially the slow release of glucose that may be beneficial to diabetic patients.

Finally, the obesity studies demonstrate a mean weight reduction of 3.9 percent within 4 weeks (Celleno et al., 2007). Clearly, if we look at the 2.93 kg weight loss by Celleno et al. [44] and the 2.24 kg loss by Wang et al. [14] over 4 weeks, then common beans may be more beneficial for weight loss compared to the other health issues we looked into. In fact, this is in concordance with the National Institute of Health [50] recommendation of 0.45 to 0.90 kg weekly weight loss that may not be detrimental to human health. Common beans are evidently important in weight loss, but again, the non-standardized bean varieties and form of bean administered may pose challenges for nutritional recommendations. Unless a standardized measure in terms of the form of bean, the number of times consumed, the quantity consumed, and the variety used (among others) are standardized, it would be hard to indicate the effect of beans on health.

Future studies should focus on standardization of measurements for common beans to be consumed and an increased number of participants, in order to provide recommendations for public nutrition policies that target the use of common beans as an alternative component in a healthy diet. Currently, studies have not yet achieved this.

## 5. Limitations

Our analysis faced challenges similar to Bazzano et al. [25], especially the fact that very few studies have evaluated common beans and their effect on health using human subjects. While the information for bean intake was available, measurements were not standardized, and common bean varieties differed significantly from study to study. It was not surprising that measurement error was inevitable for diabetes and obesity studies. Furthermore, the lack of standardized measures for cancers and CVD meant that mean reduction could not be observed, and group comparison became impossible or cumbersome. Similarly, we faced limitations in terms of sufficient data to conduct a dose–response meta-analysis, just like Zhu et al. [32]. It is possible that other unreported confounders affected the participants and thus the difference in mean outcomes. These confounders may be socio-economic and psychologically determined, but the authors did not consider them. Nevertheless, the use of covariates such as age and sex may still partially reveal the effect of beans on the sampled health variables. Moreover, our study did not include observational studies that could potentially provide a retrospective view as argued by Faraoni and Schaefer [51]. However, observational studies are a bit cumbersome to measure—for example, the postprandial glucose level after 60 minutes. Our intention was to get measurable and repeatable analyses that would be devoid of the opinions, biases, and judgements that characterize observational studies. 

## Figures and Tables

**Figure 1 nutrients-13-03701-f001:**
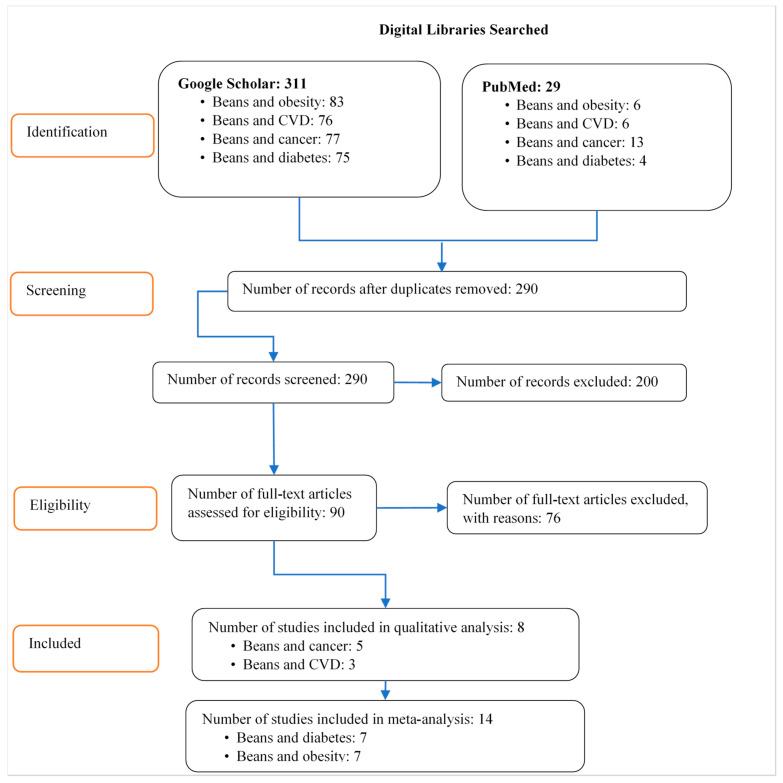
Flow chart indicating studies included in the systematic review and meta-analysis. Source: Author’s representation based on the PRISMA guidelines of Moher et al. [24].

**Figure 2 nutrients-13-03701-f002:**
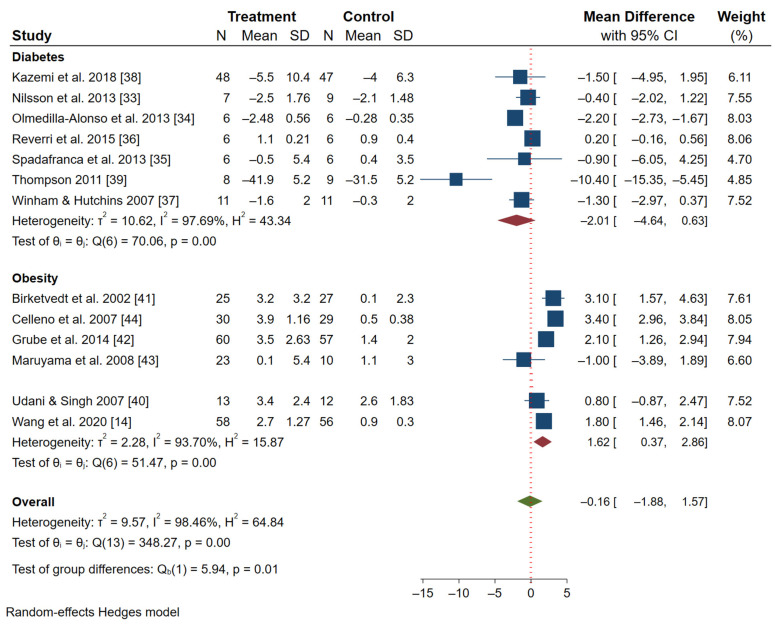
Forest plot empirical estimation of study impact for diabetes and obesity health issues.

**Table 1 nutrients-13-03701-t001:** Selected studies that characterize the association between common bean consumption and health.

Study	Study Type	Health Issue	Study Subjects	M	F	Control	Treatment	Age	Wks	Qty	Bean Form	Variety	Study Outcome(s)
Bazzano et al., 2001 [25]	Cohort	CVD	Healthy	3493	5685	-	-	25–74	12	98.6 g/day	grain	pinto, red	Beans lower risk of CVD by 11 percent.
Cryne et al., 2012 [26]	Randomized crossover	CVD	Hypercholesterolemic adults	21	21	-	-	19–40	4	100 g/day	grain	navy, pinto	Bean consumption does not affect serum lipids, homocysteine, or glycemic parameters.
Winham et al., 2007 [27]	Randomized crossover	CVD	Hypercholesterolemic adult men	7	9	-	-	22–63	8	180 g/day	grain	pinto	Serum total cholesterol decreased by 19 ± 5 mean.
Hartman et al., 2010 [28]	Randomized crossover	Cancer	Adenomas	64	0	-	-	35–75	4	251 g/day	grain	navy, pinto, kidney, black	sTNFRI/II concentrations increased by 23.8 percent.
Perera et al., 2015 [29]	Randomized crossover	Cancer	Non-smoking males	46	0	-	-	35–75	4	250 g/day	grain	navy	Serum pipecolic acid and S-methyl cysteine increased.
Baxter et al., 2019 [30]	Cohort	Cancer	Overweight and obese CRC survivors	6	13	-	-	60–65	4	35 g/day	grain	navy	The relative stool abundance of ophthalmate increased 5.25-fold for navy bean, indicating glutathione regulation.
Borresen et al., 2016 [31]	Cohort	Cancer	Colorectal cancer (CRC) survivors	12	17	-	-	59–64	4	35 g/day	grain	navy	SAA levels at week 4 improved levels associated with CRC chemoprevention.
Zhao et al., 2009 [32]	Randomized crossover	Cancer	Adenomas	23	0	-	-	35–75	4	250 g/day	grain	navy, kidney, pinto	Gene products (RNA) isolated from a stool after bean consumption had diagnostic value in assessing colon cancer risk.
Nilsson et al., 2013 [33]	Randomized crossover	Diabetes	Healthy young adults	6	19	−2.1	−2.5	23.8	2	101 g/day	grain	brown	Brown beans lowered blood glucose by 215 percent and insulin by 216 percent.
Olmedilla-Alonso et al., 2013 [34]	Randomized crossover	Diabetes	Type 2 diabetics	7	5	−0.28	−2.48	50–76	0.3	275 g/day	grain	white, cream	Only white ‘*Almonga*’ rendered a significant reduction in the triglyceridemic response.
Spadafranca et al., 2013 [35]	Randomized, double blind	Diabetes	Normal weight	6	6	0.4	−0.5	20–26	0.2	100 mg/day	extract	navy, pinto	PVE lowered postprandial glucose +15·4%, insulin +981, and C-peptide excursions in 30 min.
Reverri et al., 2015 [36]	Randomized crossover	Diabetes	Metabolic syndrome (MetS)	6	6	0.9	1.1	35–63	1.2	-	extract	black	Meals with black beans reduced postprandial insulin concentrations.
Winham and Hutchins, 2007 [37]	Randomized crossover	Diabetes	Diabetics	10	12	−0.3	−1.6	24–67	8	180 g/day	grain	navy	Total cholesterol serum (TC) for baked beans was −5.6 ± 1.5 percent
Kazemi et al., 2018 [38]	Cohort	Diabetes	Polycystic ovary syndrome (PCOS)	0	95	−4	−5.5	18–35	16	225 g/day	grain	pinto, black, kidney	The total area under the curve reduced for insulin response to a 75 g oral glucose tolerance test.
Thompson, 2011 [39]	Cohort	Diabetes	Type 2 diabetics	9	8	−31.5	−41.9	35–70	24	50 g/day	grain	pinto, black, dark red kidney	Glucose lowered for pinto, black, and red bean (compared to control) at 90, 120, and 150 min post-treatment.
Udani and Singh, 2007 [40].	Randomized, double blind	Obesity	Obese	17	8	2.6%	3.4%	18–40	4	2000 mg/day	extract	white kidney	Weight decreased by 6.0 lbs and waist size decreased by 2.2 inches.
Wang et al., 2020 [14]	Randomized, double blind	Obesity	Obese	29	27	0.9%	2.7%	18–65	4.5	2400 mg/day	extract	white kidney	Weight decreased by 2.24 kg (an average of 0.448 kg per week).
Birketvedt et al., 2002 [41]	Randomized, double blind	Obesity	Overweight and obese volunteers	21	31	0.1%	3.2%	22–66	12	900 mg/day	extract	white kidney	Serum cholesterol decreased by 6 percent in the supplement group.
Grube et al., 2014 [42]	Randomized, double blind	Obesity	BMI between 25 and 35 kg/m^2^	13	87	1.4%	3.5%	18–60	12	500 mg/day	extract	white kidney	The IQP-PV-101 group lost a mean of 2.91 ± 62.63 kg in weight.
Maruyama et al., 2008 [43]	Randomized, double blind	Obesity	Healthy women	0	33	+1.1%	0.2%	21.3	6.7	750 g/day	extract	adzuki	Triglyceride concentrations in the adzuki group decreased by 0.170 mmol/liter (15.4%).
Celleno et al., 2007 [44]	Randomized, double blind	Obesity	Overweight	17	42	0.5%	3.9%	20–45	4	445 mg/day	extract	white kidney	Weight decreased by 2.93 kg and waist circumference decreased by 4.8 cm.
Winham and Hutchins, 2007 [37]	Randomized crossover	Obesity	Healthy	10	12	-	-	24–67	8	180 g/day	grain	navy	Serum LDL-C decreased by −5.4 ± 2.3 percent.

Note: CVD—cardiovascular disease; M—male; F—female; Wks—weeks; Qty—quantity of whole bean/extract given; Ctrl—control; g/day—grams per day; sTNFRI/II—tumor necrosis factor-alpha (TNF-alpha) and its two soluble receptors, sTNFRI and sTNFRII; (SAA)—serum amyloid A protein; RNA—ribonucleic acid; PVE—*Phaseolus vulgaris* extract; IQP—PV-101 (marketed globally under the Phase 2, Starchlite, and PhaseLite brands and contains extracts of *Phaseolus vulgaris*); LDL-C—low-density lipoprotein cholesterol.

**Table 2 nutrients-13-03701-t002:** Meta-regression with mean differences as dependent variables and the moderators as predictors.

	Obesity						Diabetes					
Moderators	β	SE	t	*p* > z	95% Conf. Interval		β	SE	Z	*p* > z	95% Conf. Interval	
Age	0.19	0.05	3.50	0.07	−0.04	0.42	−0.13	0.05	−2.56	0.13	−0.35	0.09
Number of weeks	−0.26	0.09	−2.97	0.10	−0.65	0.12	−0.19	0.09	−2.07	0.17	−0.59	0.21
Quantity of beans	−0.001	0.001	−4.22	0.05	−0.003	0.001	0.02	0.01	1.72	0.23	−0.03	0.07
Constant	−1.14	1.53	−0.75	0.53	−7.74	5.45	1.20	2.02	0.60	0.61	−7.47	9.87

## Data Availability

Data for the analysis may be presented upon request using the email addresses.

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
