# Peer review of "Do Common Beans (Phaseolus vulgaris L.) Promote Good Health in Humans? A Systematic Review and Meta-Analysis of Clinical and Randomized Controlled Trials"

_nutrients, 2021, doi:10.3390/nu13113701_

Round 1

Reviewer 1 Report

The paper presents a meta-analysis on the effects of bean consumption on four health issues. It was well written and presents findings very clearly and well organized.

While there are certain limitations, such as the relatively small number of analyzed publications/data, and inconsistent standardization measures across publications, authors make an excellent effort to identify them and present them clearly with the rest of the content.

A final proof-reading is required as there are some few grammatical mistakes/errors.

Author Response

We thank you for your very important review of our paper and your endeavor to point out some oversights that as authors we couldn’t potentially see. We respond as follows to every query.

Responses to Reviewer 1

Reviewer’s comments

Response

A final proof-reading is required as there are some few grammatical mistakes/errors.

We have addressed all grammatical errors and very thankful for your review.

Reviewer 2 Report

Title: Do common beans (Phaseolus vulgaris L.) promote good health 2 in humans? A systematic review and meta-analysis of clinical 3 and randomized controlled trials

This systematic review and meta-analysis synthesized results of studies on the effect of common bean consumption on human health. The article covers important aspects of human nutrition, as these food products are an inexpensive source of protein, starch, dietary fiber and other nutrients. However, there are several major concerns that limit the validity of this work, including the writing style and use of English. Please address the following issues:

  • the reliability of the synthesized evidence depends highly on the methodological quality of a systematic review and meta-analysis. In my opinion the methodology was not described in details, especially statistical methods.
  • only randomized controlled trials and clinical trials were included in the meta-analysis. However randomized controlled trials in nutrition are limited. Therefore, the use of data from prospective, observational studies should be also considered.
  • some inappropriate terminologies were used (e.g bad cholesterol instead of LDL-cholesterol, common beans users – instead of consumers)
  • table 1 is not comprehensible, table 2 is not necessary and table 3 requires proofreading.
  • in the discussion section, the authors do not explain comprehensively the research results and mechanisms explaining the impact of common beans consumption on human health.

Author Response

We thank you for your very important review of our paper and your endeavor to point out some oversights that as authors we couldn’t potentially see. We respond as follows to every query.

Responses to Reviewer 2

Reviewer’s comments

Response

The reliability of the synthesized evidence depends highly on the methodological quality of a systematic review and meta-analysis. In my opinion the methodology was not described in details, especially statistical methods.

The sections have been re-arranged and a section describing each statistical test applied added. Specifically, we have described why we combined forest plot analysis (used in almost all meta-analyses) with meta-regression using the Knapp-Hartung error adjustment found in the Inthout et al. (2014).

Only randomized controlled trials and clinical trials were included in the meta-analysis. However randomized controlled trials in nutrition are limited. Therefore, the use of data from prospective, observational studies should be also considered.

While we are aware of the potential benefits of combining meta-analysis with observational studies as discussed in David Faraoni1 and Simon Thomas Schaefer (2016), we failed to find reliable observational studies that link common beans to health. Also, note that in observational studies, it would be a bit cumbersome to measure, for example, post-prandial glucose level after minutes. Our intention was to get measurable and repeatable analyses that would be devoid of opinions, biases and judgements that are characteristics of observational studies. In section 5 on limitations, we have included this with arguments of not including observational studies. Nonetheless, we appreciate you pointing this out and would consider using it in the future.

Some inappropriate terminologies were used (e.g bad cholesterol instead of LDL-cholesterol, common beans users – instead of consumers)

The terms have been appropriately replaced. bad cholesterol has been replaced with LDL-cholesterol and the common beans users replaced with common bean consumers

Table 1 is not comprehensible, table 2 is not necessary and table 3 requires proofreading.

Table 1 is very critical for anyone interested in knowing the type of population studied and the ranges of parameters tested. For example, if it is removed, one may ask what the age range of the subjects. Were they males or females, how many weeks did the intervention occur etc? Without such information, it would be impossible to conclude, for example, that weekly bean consumption help in weight loss. Table 2 is necessary because the organization that sponsored the study promotes bean production, sales and consumption. Specifically, varietal preferences would inform the potential varieties to promote health benefits. This table also indicates that some varieties are more preferred in RCTs than others and the need to change or try other varieties. Table 3 shows summarized form of the studies that were included in the meta-analysis and their findings so that one may need to read them further. Almost all meta-analysis studies tend to report such table.

In the discussion section, the authors do not explain comprehensively the research results and mechanisms explaining the impact of common beans consumption on human health.

Additional explanation of results in discussion section has been added.

Round 2

Reviewer 2 Report

Thank you to the authors for responding to the comments, however, I still have some remarks.

  • I still maintain that table 1 is not comprehensible, table 2 is not necessary and table 3 requires proofreading. Author's reply to the answer is not satisfactory: Table 1 is very critical for anyone interested in knowing the type of population studied and the ranges of parameters tested. For example, if it is removed, one may ask what the age range of the subjects. Were they males or females, how many weeks did the intervention occur etc?- such information is given in table 3. What is the point of reporting the average sample size if the range of study individuals varies from 12 people to over 9,000? I suggest removing tables 1 and 2, and improving table 3. The latter is difficult to interpret (rows are shifted). I also suggest to add columns: the type of study, study main outcomes (Instead of summary). It is important to clearly indicate which studies were conducted on healthy people and which on people with diseases.
  • Inclusion criteria defined by authors are among others (line 111): (4) used RCT or clinical trials. However some studies included in this meta-analysis do not meet these criteria e.g.:

Tucker, L. A. (2020). Bean Consumption Accounts for Differences in Body Fat and Waist Circumference: A Cross-Sectional Study of 246 Women. 294  Journal of Nutrition and Metabolism, 2020. https://doi.org/10.1155/2020/9140907. Please check whether all items meet the criteria.

Author Response

Responses to reviewer 2

  • I still maintain that table 1 is not comprehensible, table 2 is not necessary and table 3 requires proofreading. Author's reply to the answer is not satisfactory: Table 1 is very critical for anyone interested in knowing the type of population studied and the ranges of parameters tested. For example, if it is removed, one may ask what the age range of the subjects. Were they males or females, how many weeks did the intervention occur etc?- such information is given in table 3. What is the point of reporting the average sample size if the range of study individuals varies from 12 people to over 9,000? I suggest removing tables 1 and 2, and improving table 3. The latter is difficult to interpret (rows are shifted). I also suggest to add columns: the type of study, study main outcomes (Instead of summary). It is important to clearly indicate which studies were conducted on healthy people and which on people with diseases.

Response: We have removed tables 1 and 2 and edited Table 3 to be the first table. The additional columns added include study type and study subjects. We have re-arranged the columns to flow as suggested. We have, however noted with concern, the cluttering of Table 3 having integrated the reviewer’s suggestions.

  • Inclusion criteria defined by authors are among others (line 111): (4) used RCT or clinical trials. However some studies included in this meta-analysis do not meet these criteria e.g.: Tucker, L. A. (2020). Bean Consumption Accounts for Differences in Body Fat and Waist Circumference: A Cross-Sectional Study of 246 Women. 294  Journal of Nutrition and Metabolism, 2020. https://doi.org/10.1155/2020/9140907. Please check whether all items meet the criteria.

Response: We have critically reviewed the paper and purged it from our list including all sections associated with it. We thanks you for pointing it out.
